# Mitigating Trunk Compensatory Movements in Post-Stroke Survivors through Visual Feedback during Robotic-Assisted Arm Reaching Exercises

**DOI:** 10.3390/s24113331

**Published:** 2024-05-23

**Authors:** Seong-Hoon Lee, Won-Kyung Song

**Affiliations:** Department of Rehabilitative and Assistive Technology, National Rehabilitation Center, Seoul 01022, Republic of Korea; lsh1799@korea.kr

**Keywords:** arm reaching, stroke, rehabilitation, visual feedback, robot, trunk compensation

## Abstract

Trunk compensatory movements frequently manifest during robotic-assisted arm reaching exercises for upper limb rehabilitation following a stroke, potentially impeding functional recovery. These aberrant movements are prevalent among stroke survivors and can hinder their progress in rehabilitation, making it crucial to address this issue. This study evaluated the efficacy of visual feedback, facilitated by an RGB-D camera, in reducing trunk compensation. In total, 17 able-bodied individuals and 18 stroke survivors performed reaching tasks under unrestricted trunk conditions and visual feedback conditions. In the visual feedback modalities, the target position was synchronized with trunk movement at ratios where the target moved at the same speed, double, and triple the trunk’s motion speed, providing real-time feedback to the participants. Notably, trunk compensatory movements were significantly diminished when the target moved at the same speed and double the trunk’s motion speed. Furthermore, these conditions exhibited an increase in the task completion time and perceived exertion among stroke survivors. This outcome suggests that visual feedback effectively heightened the task difficulty, thereby discouraging unnecessary trunk motion. The findings underscore the pivotal role of customized visual feedback in correcting aberrant upper limb movements among stroke survivors, potentially contributing to the advancement of robotic-assisted rehabilitation strategies. These insights advocate for the integration of visual feedback into rehabilitation exercises, highlighting its potential to foster more effective recovery pathways for post-stroke individuals by minimizing undesired compensatory motions.

## 1. Introduction

Arm reaching tasks are crucial in the rehabilitation of the upper limb, providing significant therapeutic benefits for individuals recovering from strokes [1]. However, these exercises often lead to trunk compensation, where trunk movements compensate for deficits in upper limb strength and coordination. This compensatory strategy can hinder rehabilitation and increase the risk of secondary complications [2,3]. Given the complex nature of motor learning mechanisms and their impact on rehabilitation, integrating these insights into therapy is essential for optimizing outcomes [4]. Thus, developing effective strategies to mitigate trunk compensation is critical [5].

Visual feedback has been recognized as an effective method for enhancing motor skill learning and rehabilitation outcomes [6,7,8]. Research has shown the benefits of visual feedback techniques, such as mirror therapy and video feedback, in correcting abnormal movement patterns among stroke survivors. Furthermore, virtual reality-based visual feedback has the potential to improve upper limb function. However, most studies have focused on incorporating visual feedback into traditional physical therapy, with less attention to robotic-assisted rehabilitation. This study aims to explore the use of real-time visual feedback in reducing trunk compensatory movements within a robotic-assisted rehabilitation setting, expanding the scope of previous research [9]. We have developed a user-friendly visual feedback system using an RGB-D camera, eliminating the need for body-mounted sensors.

Traditionally, trunk compensation has been addressed by physically restraining the user’s trunk with straps, leading to discomfort and adaptability challenges, especially in unsupervised home settings [10]. The introduction of robotic-assisted reaching exercises using end-effector technology marks a significant advancement in upper limb rehabilitation [11,12]. This approach offers precise control over arm movements and integrates virtual reality and various feedback methods to facilitate motor learning. However, the effectiveness of feedback techniques in reducing trunk compensation in this context remains underexplored [9].

Extensive research in rehabilitation science has focused on trunk compensation during arm reaching tasks [2,10,13]. Studies report that excessive trunk movement, common among those with upper limb impairments, negatively impacts rehabilitation outcomes [10,14]. To combat this, feedback-based strategies, including visual, auditory, and multimodal feedback, have been proposed [6,15,16].

Visual feedback methods provide instant visual cues about trunk posture and movement, enabling real-time corrections [6,7,8,17,18]. Auditory feedback uses sound signals to alert deviations from optimal trunk alignment [19,20,21], while multimodal feedback combines visual, auditory, and kinesthetic cues to reinforce motor learning and minimize trunk compensation [6,7,17].

Despite the potential of robotic-assisted reaching exercises in enhancing rehabilitation and targeting trunk compensation, challenges remain, such as inconsistent results due to varied environmental conditions and a lack of comprehensive feedback modality evaluations [5,22,23,24,25]. As the use of home-based rehabilitation robots by stroke survivors increases, addressing trunk compensation is imperative [26]. This has led to exploring innovative feedback technologies, like real-time adjustments with RGB-D cameras, and avoiding external devices’ need.

This study investigates trunk compensation in end-effector robot-assisted reaching exercises and assesses the effectiveness of visual feedback techniques in mitigating this strategy [13,26]. By utilizing visual feedback without body-mounted sensors, this approach is advantageous for home-based rehabilitation. Furthermore, by comparing the performance of able-bodied individuals and stroke survivors, this study seeks to identify effective interventions for enhancing upper limb function and pioneering the practical impact of using robots and visual feedback to suppress trunk compensation during reaching tasks. This focus on real-world outcomes for individuals with disabilities contributes significantly to the evolving field of rehabilitation, emphasizing the translational potential of these findings in optimizing rehabilitation outcomes.

## 2. Materials and Methods

This section provides an overview of the materials and methods employed in the study, which investigated the efficacy of visual feedback modalities in reducing trunk compensatory movements during robotic-assisted arm reaching exercises in post-stroke individuals. The study consisted of five main components: (1) the introduction of the NREH device, a novel robotic rehabilitation system designed for home use by stroke survivors; (2) the detection of trunk motion using an RGB-D camera, leveraging advanced computer vision techniques; (3) participant recruitment and demographics, including both able-bodied individuals and stroke survivors; and (4) experimental setup and conditions, detailing the visual feedback modalities and reaching task parameters. Additionally, (5) the data analysis and statistical methods used to evaluate the participants’ performance and the effectiveness of the visual feedback are described.

### 2.1. NREH: NRC End-Effector-Based Rehabilitation Arm at Home

The National Rehabilitation Center has introduced the NREH [27,28], an innovative rehabilitation device engineered for home use by stroke survivors (Figure 1). Surpassing the capabilities of conventional wearable exoskeletons such as the NREX, the NREH features a motorized arm capable of executing a wide range of motions, complemented by a handgrip designed to enhance patient interaction. This device uniquely adapts rehabilitation exercises to accommodate the evolving needs of patients, offering adjustable difficulty levels and targeted feedback to optimize therapeutic outcomes.

The NREH operates on a 5-bar linkage system, primarily enabling movements in a 2D plane. While it possesses the capacity to simulate 3D movements through workspace tilting, the present study maintained a flat configuration to focus on evaluating the influence of visual feedback and robotic application on trunk movement during reaching tasks. Powered by Robotis smart actuators, the device ensures precise arm control. Furthermore, its software facilitates customized feedback and data acquisition, enabling a personalized rehabilitation experience. The incorporated handle supports individuals with upper limb impairments in safely executing exercises, thereby promoting independence and recovery in a home setting. This approach highlights the versatility of the NREH and its role in advancing upper limb rehabilitation by specifically targeting trunk movements in a simplified setup.

### 2.2. Detecting Trunk Motion via RGB-D Camera

We leveraged the capabilities of Microsoft’s Azure Kinect (Redmond, WA, USA) and its Body Tracking SDK 1.1.2 for accurate trunk motion detection, employing the NFOV (Narrow Field of View) depth mode in conjunction with an RTX3080 GPU (NVIDIA, Santa Clara, CA, USA) to ensure high precision [29]. The system architecture facilitated seamless integration between hardware and software components, with the NREH control program developed in C++ and visual feedback mechanisms implemented in C# within the Unity framework, resulting in a dynamic and interactive rehabilitation experience (Figure 2). Communication between the control and feedback systems was established through TCP/IP socket communication, streamlining the exchange of data, and including the robotic handgrip coordinates and trunk movement measurements. This setup enabled a real-time visual representation of the exercises, as depicted in the block diagram of the NREH system in Figure 2.

Unity game target adjustments were based on the incoming trunk coordinate data, enabling adaptive exercise difficulty tailored to the patient’s performance. The trunk position was determined by averaging the coordinates of three anatomical landmarks: the left and right clavicles and the sternum, as described in Equation (1), where ***P****_Trunk_*, ***P****_CL_*, ***P****_CR_*, and ***P****_SC_* represent the Cartesian coordinates of these key points, respectively (Figure 3). This methodology emphasized the integration of advanced tracking technology with interactive software, streamlining the process of capturing and responding to patient movements to promote a more engaging and effective rehabilitation process.
***P****_Trunk_* = (***P****_CL_* + ***P****_CR_* + ***P****_SC_*)/3(1)

When delivering real-time visual feedback using an RGB-D camera, it is crucial to consider vibrations and nonlinear factors. To address these issues, hysteresis comparators and moving average filters were applied, altering the output signal when the input exceeded a certain threshold. In this study, the threshold was set to approximately three times the standard deviation of the trunk’s RGB-D camera measurement (Figure 4). Although Azure Kinect can synchronize up to eight devices to enhance accuracy, it demands ample space for setup. To facilitate trunk data collection for home use, this research utilized a single Azure Kinect.

### 2.3. Experimental Setup

Participants were seated in a stationary chair, maintaining a stable posture with their feet on the ground and their back against the chair. The Azure Kinect camera was strategically positioned to capture only the participant’s movements, with a curtain shielding potential visual distractions. Each participant engaged in reaching exercises using the NREH robot, utilizing their affected arm for stroke survivors or nondominant arm for able-bodied individuals.

In this study, visual feedback was delivered using the Unity game engine. The target position within Unity was adjusted in real-time based on the trunk coordinate data received from the RGB-D camera. Participants performed the task of reaching the moving target by manipulating the robotic arm. As the target was synchronized with trunk movements, excessive trunk motion made it challenging to reach the target. This approach allowed participants to execute the arm reaching task while minimizing trunk movements based on visual cues. Similar methods of providing visual feedback have been employed in previous studies, such as Valdés et al. (2017) [6] and Cai et al. (2020) [9], who successfully utilized mirror feedback and virtual reality feedback, respectively, to reduce compensatory movements in stroke patients. Building upon these prior studies, the current research distinguishes itself by promoting participants’ active utilization of feedback through real-time target manipulation.

The exercises involved reaching towards a target positioned at desk level, with the arm maintained at a 90-degree elbow flexion.

The study categorized the reaching exercises into six distinct conditions to assess varying levels of task complexity and trunk involvement:
Condition #1: Reaching with fixed trunk (trunk restraint)Condition #2: Reaching with free trunk (free trunk)Condition #3: Free trunk with a synchronously moving target
***P****_target_* = ***P****_target_* + Δ***P****_Trunk_*(2)Condition #4: Free trunk with the target moving at twice the speed of trunk movement
***P****_target_* = ***P****_target_* + 2Δ***P****_Trunk_*(3)Condition #5: Free trunk with the target moving at three times the speed of trunk movement
***P****_target_* = ***P****_target_* + 3Δ***P****_Trunk_*(4)Condition #6: Free trunk with the target moving at twelve times the speed of trunk movement (primarily to challenge able-bodied participants; considered unsuitable for stroke survivors due to high difficulty)
***P****_target_* = ***P****_target_* + 12Δ***P****_Trunk_*(5)

In these equations, ***P****_target_* represents the target position, which was updated by adding the change in trunk position Δ***P****_Trunk_*, multiplied by the respective speed factor (1, 2, 3, or 12), to the current target position. The ratios of 1:1, 1:2, and 1:3 for synchronizing the target position with trunk movement were selected to provide visually perceptible target displacements for reducing trunk compensation. The 1:1 ratio represents the most basic level of feedback, where the target position precisely reflects trunk motion. The 1:2 and 1:3 ratios progressively increase difficulty to elicit greater reductions in trunk movement. Preliminary experiments indicated that ratios exceeding 1:3 became excessively challenging even for able-bodied participants and were thus excluded from the main study.

By implementing these incremental ratios, we could systematically evaluate the effects of the visual feedback. Lower ratios allowed us to examine the intuitiveness of the feedback, while higher ratios revealed the impact of task difficulty modulation. Ultimately, this study aimed to identify the suitable level of visual feedback for robotic-assisted rehabilitation.

Applying visual feedback [31] in the reaching task, it was hypothesized that increasing trunk compensation would subsequently elevate the task’s difficulty by moving the target further away. This methodology, translating subtle trunk movements into visually perceptible feedback, aimed to reduce trunk compensation by making reaching the target more challenging.

Targets were displayed in three directions (medial, contralateral, and ipsilateral) relative to the participant, positioned at 100% of their maximum comfortable reaching distance, facilitating a point-to-point reaching task without robotic guidance. Participants completed a series of 40 reaching trials per target direction under each condition, culminating in 120 reaches per condition. The order of target presentation was consistent across all conditions, starting with the central target and progressing to the right and left targets. To minimize the influence of sequential effects on the experimental results, appropriate rest periods were provided between conditions. The tasks were presented in order of increasing difficulty, starting with the easiest task and progressing to more challenging tasks, to maintain participants’ motivation throughout the experiment. Participants performed between 600 and 720 total reaches, with data collection conducted at a sampling rate of approximately 30 Hz. Trunk movement data were captured via the Azure Kinect RGB-D camera, while the NREH robot facilitated the exercises and data processing, ensuring a comprehensive analysis of each participant’s performance during the reaching tasks.

### 2.4. Participants

This study was approved by the Institutional Review Board (IRB) of the National Rehabilitation Center (IRB #: NRC-2022-07-057), and all participants provided written informed consent prior to their participation.

This study included 17 able-bodied individuals (5 males, 12 females) with no history of upper limb injuries or neurological conditions, alongside 18 stroke survivors (14 males, 4 females) exhibiting impaired reaching abilities. The sample size was determined based on similar studies in the field [32,33], and participants were recruited through convenience sampling at the National Rehabilitation Center.

The inclusion criteria for stroke survivors were as follows: (1) aged 19 years or older, (2) diagnosed with a unilateral stroke resulting in hemiplegia, (3) at least 6 months post-stroke, (4) Medical Research Council (MRC)_shoulder score ≥ 2, (5) Modified Ashworth Scale (MAS) score < 3, (6) Line Bisection Test (LBT) result < 12.5 mm, and (7) Korean Mini-Mental Status Examination (K-MMSE) score > 18. Exclusion criteria included the following: (1) visual field defects or reduced visual acuity that would prevent the participant from perceiving changes on the screen, (2) neurological or orthopedic problems in the upper limbs (instability, dislocation, fracture, etc.), (3) currently receiving medication for pain, and (4) severe contracture in the elbow joint that would prevent movement.

The able-bodied group, consisting of 17 participants, and 16 out of 18 stroke survivors successfully fulfilled the task requirements, with incomplete or missing data excluded from the final analysis. Five able-bodied participants were excluded from Condition #5 due to technical issues encountered during the setup.

The mean age of the able-bodied group was 34.23 years (SD = 6.63), contrasting with the stroke survivor group, which had a mean age of 64.16 years (SD = 11.03). Detailed demographic and clinical characteristics of the stroke group included an average duration of 13.5 years (SD = 8.06) since stroke onset, Fugl-Meyer Assessment (FMA) scores averaging 41.67 (SD = 15.38), Box and Block Test (BBT) results of 11.67 (SD = 12.35), Action Research Arm Test (ARAT) scores of 29.5 (SD = 21.81), and Modified Barthel Index (MBI) scores of 88.56 (SD = 20.78), providing a comprehensive overview of their functional abilities and rehabilitation potential.

### 2.5. Data Analysis and Statistical Analysis

Trunk motion plays a crucial role in arm reaching tasks, and specific indicators were employed to assess the extent and efficiency of trunk movement. Utilizing SPSS software (Version 19; IBM Corp., Armonk, NY, USA) and the Wilcoxon signed-rank test, a comprehensive analysis of the data was conducted across all participants who successfully completed the exercises under each condition. The able-bodied group, consisting of 17 participants, and 16 out of 18 stroke survivors successfully fulfilled the task requirements, with incomplete or missing data excluded from the final analysis. Five able-bodied participants were excluded from Condition #5 due to technical issues encountered during the setup.

The evaluation of trunk motion focused on the following key aspects:
Trunk Range of Motion: This essential metric quantified the magnitude of trunk displacement during the task, with a larger range indicating more extensive trunk movement. However, for optimal stability and task efficiency, a constrained range of motion was desirable.Trunk Position Difference: Assessing the difference between the trunk’s initial and final positions provided valuable insights into the task’s stability and efficiency. A smaller difference suggested consistent trunk control throughout the reaching movement, which was considered ideal.Movement Time: The time required to complete 120 reaching attempts under various conditions was recorded, offering a measure of task engagement duration.Borg Scale: The Borg scale [34], ranging from 6 to 20, was employed to assess participants’ perceived effort level for each condition. Higher scores indicated greater perceived exertion or difficulty experienced by participants during the task.

These parameters provided a comprehensive understanding of trunk movement’s impact on the arm reaching task, emphasizing the significance of minimizing unnecessary trunk motion to optimize task performance. This approach facilitated the refinement of rehabilitation exercises, enhancing their effectiveness for both able-bodied individuals and stroke survivors.

## 3. Results

This section presents the results of the study, focusing on the evaluation of trunk motion and the effectiveness of visual feedback in reducing trunk compensatory movements during robotic-assisted arm reaching exercises. The analysis is divided into five main subsections: (1) trunk range of motion, (2) trunk position difference between the starting and ending positions, (3) movement time, (4) perceived effort, and (5) trajectory. The results are presented using descriptive statistics, box plots, and statistical tests to highlight significant differences between conditions and groups, providing insights into the implications for visual feedback strategies in robotic rehabilitation.

### 3.1. Trunk Range of Motion

Figure 5 presents the trunk range of motion (ROM) analysis, displaying the minimum and maximum values of trunk position coordinates across different conditions, with the Wilcoxon signed-rank test determining the significance of the observed differences.

In the stroke group, significant variations were primarily identified between Condition #1 and Condition #2. The absence of significant disparities from Condition #1 through Conditions #3 to #5 suggests that scenarios incorporating trunk resistance alongside visual feedback regarding trunk motion exhibit comparable effects. Notably, Condition #4, characterized by doubled visual feedback of trunk motion, demonstrated the lowest average trunk ROM among stroke participants.

Conversely, the able-bodied group exhibited no substantial difference between the restrained (Condition #1) and unrestrained (Condition #2) states or between Conditions #3 to #5. This finding indicates that the visual feedback of trunk movement maintained a consistent pattern within the able-bodied individuals, similar to Condition #1. However, marked distinctions emerged in Condition #6, an extreme feedback condition, and when trunk motion was visually amplified more than threefold in the unrestrained state (Condition #2), suggesting an unfamiliar discomfort due to trunk restriction.

Furthermore, Figure 5 presents both average and median values, with the average representing the mean of all values, and the median, excluding outliers, representing the midpoint [35]. The stroke group’s median trunk ROM was lowest in Condition #3, while its average was minimal in Condition #4. These findings suggest that Condition #4, with target synchronization at twice the trunk motion, was optimally suited for reducing trunk movement on average. However, when considering the median and excluding outliers, Condition #3, involving target synchronization equivalent to trunk motion, also appeared to be effective.

### 3.2. Trunk Position Difference between the Starting and Ending Positions

The assessment of the trunk position difference between the starting and ending positions revealed significant findings (Figure 6). In the stroke group, a notable disparity was observed between Condition #1, characterized by restricted trunk movement, and Condition #5. Conversely, Conditions #2 to #4 exhibited trends parallel to Condition #1, indicating consistency across these conditions. Remarkably, Condition #5 demonstrated the lowest values within the stroke group, suggesting enhanced stability.

In contrast, the able-bodied group did not exhibit significant differences between Condition #1, with constrained movement, and Conditions #2 to #5, highlighting a uniform trend even when integrating the visual feedback of trunk movement. However, a distinct divergence was noted in Condition #6, an extreme feedback condition, underscoring the impact of heightened visual feedback.

Interestingly, both groups experienced an increase in trunk position difference in Condition #3 compared to Condition #2, indicating that a 1:1 synchronization of the target with trunk movement via visual feedback promoted additional trunk motion. Conversely, an inverse relationship was observed as the multiplier for target synchronization increased, resulting in a decreased trunk position difference.

Specifically, the stroke group exhibited the smallest difference in trunk position from start to end of the reach in Condition #2, primarily due to the stationary target. However, in Conditions #3 to #5, where the target moved in synchronization with the trunk, participants were required to reach further, leading to an increased difference in trunk position from start to end. Notably, Condition #4, with the target moving twice as far as the trunk, resulted in the smallest difference in trunk position, demonstrating the effectiveness of this approach in reducing unnecessary motion by directly challenging participants.

The directional analysis revealed that the able-bodied group exhibited the most significant trunk movement in the contralateral direction, while the stroke group showed predominant movement in the ipsilateral direction (Figure 7). This pattern aligns with the natural motor behaviors of each group, where able-bodied individuals tend to display larger shoulder movements during contralateral reaches, whereas stroke survivors demonstrate pronounced ipsilateral movements, reflecting the characteristic ipsilateral motor deficit commonly observed in stroke patients [26].

Across all directions, trunk movement was more pronounced in the stroke group compared to the able-bodied group, with an overall trend indicating that increasing visual feedback of trunk movement contributes to the reduction of trunk motion in every direction. This suggests a potential strategy for enhancing task stability and efficiency.

### 3.3. Movement Time

Figure 8 depicts the total time required to complete 120 reaching attempts under each condition for both the stroke and able-bodied groups. The stroke group exhibited the longest average duration for the reaching tasks in Condition #1, with the time required increasing as the synchronization between the trunk and target intensified.

Interestingly, both groups demonstrated a slight reduction in task completion time in the more challenging Condition #5, suggesting that participants might have adopted a quicker movement strategy to cope with the increasing distance of the target. However, for the able-bodied group, the use of extreme visual feedback in Condition #6 significantly extended the duration, contrasting with the other conditions.

### 3.4. Perceived Effort

The Borg scale, ranging from 6 to 20 points [34], was employed by participants to evaluate their perceived effort for each reaching task (Figure 9). After completing each task, feedback scores were collected and averaged to assess the perceived level of exertion. Figure 9 shows the average Borg scores for each condition, revealing an overall increase in perceived effort as participants progressed through the tasks.

In the stroke group, no significant difference in Borg scores was observed between Conditions #1 and #2, suggesting that the restriction of trunk movement does not significantly influence the perceived level of effort when engaging with stationary targets. However, a statistically significant difference was found across all other conditions, indicating that the application of visual feedback in tasks with a fixed target increases perceived effort. This highlights the impact of visual feedback on the participants’ perception of effort when engaging with stationary targets.

### 3.5. Trajectory

Figure 10 illustrates the trajectories of trunk movement for representative individuals from both the stroke survivor and able-bodied groups under various conditions. For the stroke survivor, a notable decrease in the trunk movement trajectory was observed in Condition #4, mirroring the pattern seen in their trunk range of motion. Conversely, the able-bodied participant exhibited a significant reduction in the trajectory, particularly evident in Condition #6, aligning with the trends observed in the trunk range of motion.

Interestingly, Condition #4 was characterized by a reduction in trunk movement trajectory accompanied by an increase in task completion time for the able-bodied group, highlighting the potential impact of visual feedback on movement efficiency and temporal performance.

## 4. Discussion

Trunk compensation is frequently employed by stroke survivors during arm reaching tasks, which can hinder functional recovery in upper limb robotic rehabilitation therapy. This study investigated the feasibility of preventing trunk compensation without a trunk fixation belt by measuring and providing visual feedback on the user’s trunk movement using an RGB-D camera. Incorporating motion capture devices such as RGB-D cameras and virtual reality technology into real-time visual feedback has proven beneficial for stroke survivors in correcting their movements.

The results of this study demonstrate that synchronizing trunk movement with the target position decreases trunk compensation while enhancing sensory feedback and improving the efficacy of trunk strengthening exercises. From the perspective of trunk range of motion (ROM), Condition #3 was found suitable for reducing trunk compensation among stroke participants. Meanwhile, considering the difference in trunk position between the starting and ending positions, Condition #4 was identified as the most effective in minimizing trunk compensation. Thus, applying target movements ranging from 1 to 2 times the trunk movement is deemed reasonable for stroke participants. These findings can be explained by the principles of sensory-motor learning and error-based learning theories. By providing real-time visual feedback on trunk movement, the system helps stroke survivors to detect and correct their movement errors, facilitating the learning of proper motor patterns and reducing reliance on compensatory strategies.

Furthermore, increasing the duration to reach a target can positively influence stroke rehabilitation by reducing trunk compensation. This study demonstrated the ability to adjust the difficulty of the reaching task by synchronizing the trunk and target movements. As the task’s difficulty increased, trunk compensation decreased, and the time to reach the target extended. However, excessively increasing the difficulty (Condition #5, with target movements three times the trunk movement) paradoxically induced trunk compensation and shortened the reaching time, highlighting the importance of appropriate task difficulty modulation.

This research proposes visual feedback techniques for suppressing trunk compensation in reaching tasks utilizing upper limb rehabilitation robots. Synchronizing the target with the trunk has been effective in reducing trunk compensation, thereby increasing the efficiency of reaching training. By measuring trunk movement with an RGB-D camera and adjusting the reaching task’s difficulty through visual feedback, this study highlights the potential to increase user engagement and improve the effectiveness of rehabilitation programs for stroke survivors.

The results of this study suggest the usefulness of visual feedback in upper limb rehabilitation for stroke patients. Noncontact motion tracking using RGB-D cameras and real-time feedback provision offer higher usability and accessibility compared to traditional marker-based systems, making them suitable for clinical applications [36]. In particular, the visual feedback system proposed in this study can be directly integrated into robot-assisted rehabilitation training, potentially contributing to the provision of personalized interventions for patients.

However, it is important to acknowledge that the present study primarily focused on the immediate effects of visual feedback on trunk compensation. Further research is necessary to determine whether the observed changes in the evaluation metrics are due to transient effects of the feedback modality or actual motor skill learning. Future experiments should be designed to evaluate stroke patients’ functional improvements and the retention and generalization ability of different visual feedback training protocols over an extended period.

Additionally, as this study only verified short-term effects, further research is needed to confirm the long-term clinical efficacy. Additionally, large-scale clinical studies involving stroke patients with various severities and types should be conducted to validate the effectiveness of the proposed visual feedback system. Despite these limitations, the results of this study are meaningful in demonstrating the potential value of visual feedback in robotic-assisted rehabilitation.

## 5. Conclusions

This study investigated the efficacy of visual feedback modalities in reducing trunk compensatory movements during robotic-assisted arm reaching exercises in post-stroke individuals. By comparing the performance of stroke survivors under various visual feedback conditions to that of a control group, we aimed to evaluate the potential of RGB-D camera-based strategies in mitigating trunk compensation. The findings underscore the significant impact of visual feedback in effectively reducing trunk compensation, highlighting its crucial role in the development of optimal therapeutic interventions for stroke rehabilitation.

The consistent results across different visual feedback conditions in the stroke group lay the foundation for future research directions. To comprehensively understand the influence of multimodal feedback on trunk compensation and functional recovery, it is essential to expand the scope of investigation to include diverse modalities such as haptic and auditory feedback. Future studies should explore the optimal combination and timing of these feedback modalities to maximize their effectiveness in reducing compensatory movements and promoting motor learning. Moreover, conducting longitudinal clinical trials is crucial to assess the long-term effects and sustainability of feedback-based interventions. These trials should include follow-up assessments at various time points to evaluate the retention and generalization of the learned motor patterns.

Future research should incorporate comparisons between groups receiving visual feedback for trunk compensation control and those without. Randomized controlled trials comparing feedback-based interventions to conventional therapy approaches would provide valuable insights into the relative efficacy and cost-effectiveness of these strategies. Outcome measures should include not only trunk compensation but also functional outcomes, quality of life, and patient satisfaction. By employing robust study designs and larger sample sizes, these investigations will contribute to the development of evidence-based guidelines for integrating visual feedback into clinical practice. These guidelines should address practical considerations such as the optimal frequency and duration of feedback sessions, the most effective feedback modalities for different stages of recovery, and strategies for tailoring feedback to individual patient needs.

In conclusion, this study establishes the promising role of visual feedback in reducing trunk compensatory movements during robotic-assisted arm reaching exercises in stroke rehabilitation. The findings emphasize the need for continued research to optimize feedback modalities, assess long-term efficacy, and compare feedback-based interventions to conventional approaches. By advancing our understanding of visual feedback in robotic rehabilitation, we can work towards developing more targeted and effective strategies to enhance functional recovery for individuals affected by stroke. Ultimately, the integration of evidence-based visual feedback strategies into clinical practice has the potential to revolutionize stroke rehabilitation, leading to improved outcomes and quality of life for stroke survivors.

## Figures and Tables

**Figure 1 sensors-24-03331-f001:**
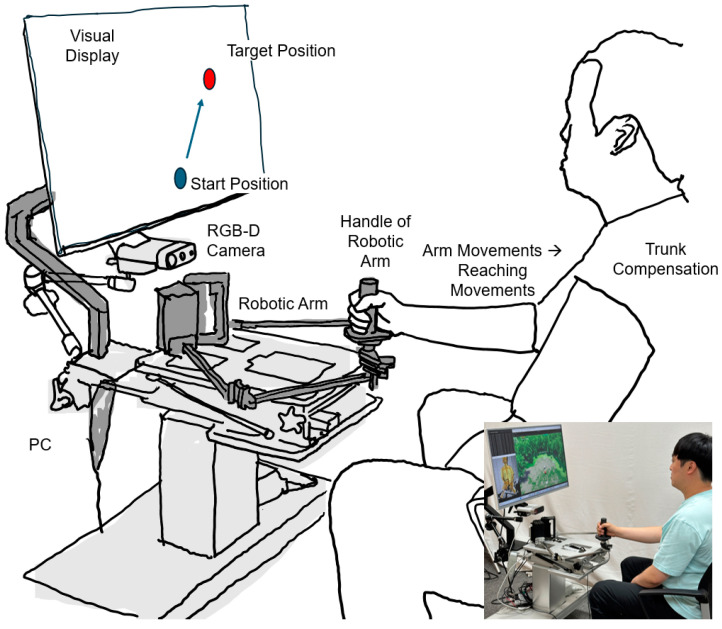
NREH: NRC end-effector-based rehabilitation arm at home.

**Figure 2 sensors-24-03331-f002:**
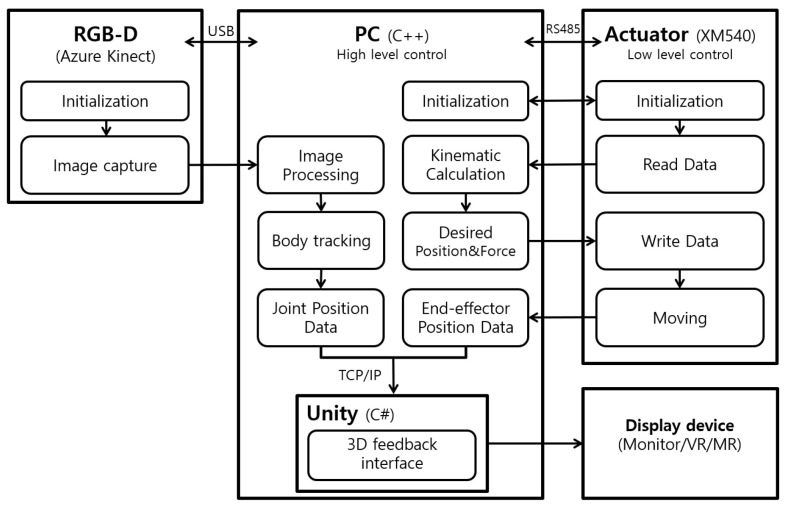
System block diagram.

**Figure 3 sensors-24-03331-f003:**
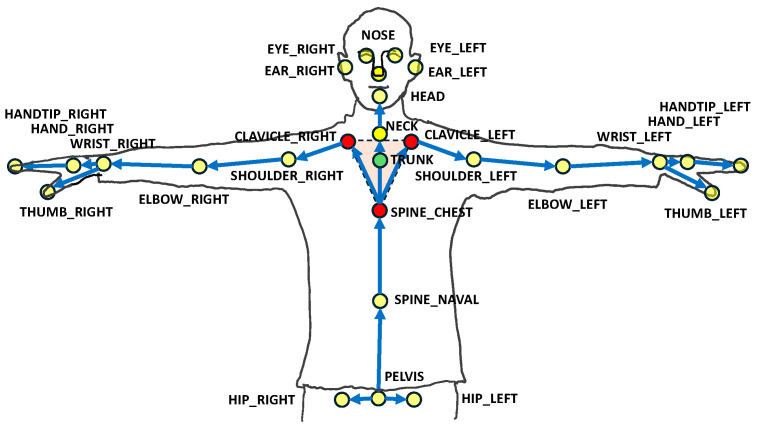
Marker positions. The movement of the trunk was determined using the average positions of three locations (CLAVICLE_LEFT, CLAVICLE_RIGHT, SPINE_CHEST) [30]. The red dots represent the positions of ***P****_CL_*, ***P****_CR_*, and ***P****_SC_*, while the green dot represents the position of ***P****_Trunk_*.

**Figure 4 sensors-24-03331-f004:**
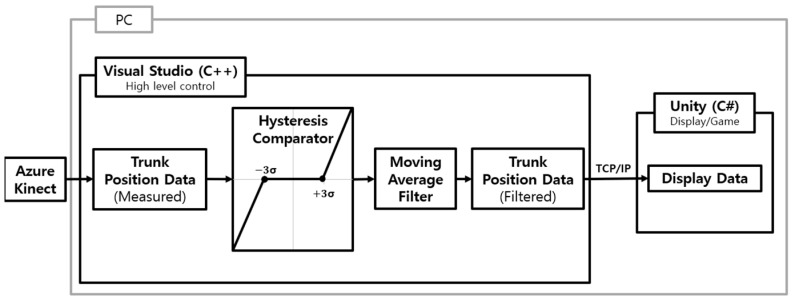
Block diagram of position data filtering process.

**Figure 5 sensors-24-03331-f005:**
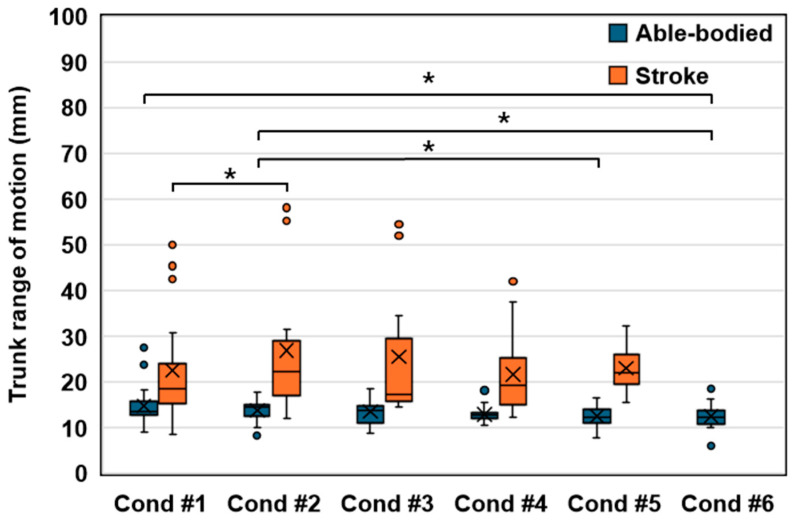
Box plot illustrating the average trunk range of motion among all participants, where X marks indicate mean values. Due to its high difficulty, the stroke group did not attempt Condition #6, and only performed Conditions #1 through #5 for successful task execution. The asterisk indicates statistical significance (*p* < 0.05).

**Figure 6 sensors-24-03331-f006:**
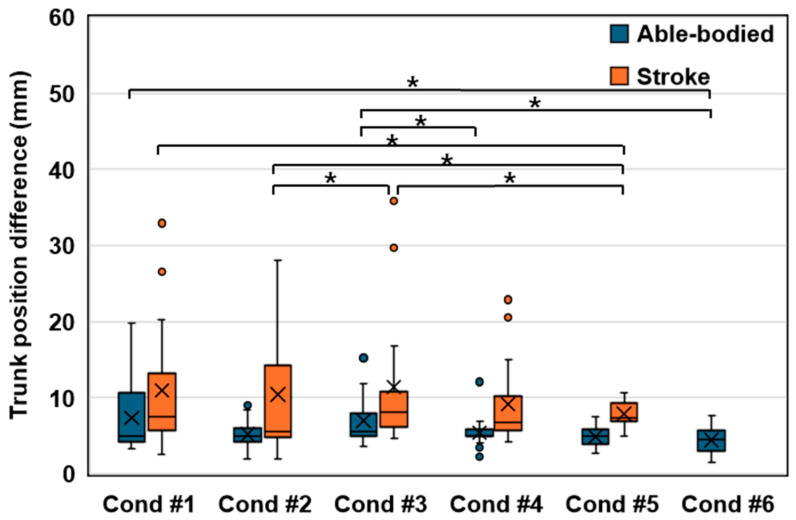
Box plot illustrating the difference in trunk position between the starting and ending positions across various conditions, where X marks indicate mean values. Due to its high difficulty, the stroke group did not attempt Condition #6, and only performed Conditions #1 through #5 for successful task execution. The asterisk indicates statistical significance (*p* < 0.05).

**Figure 7 sensors-24-03331-f007:**
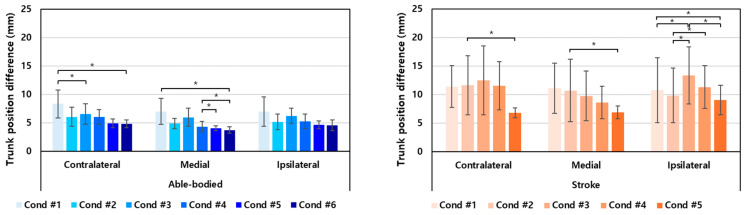
Trunk position difference between the starting and ending positions for each direction. The asterisk indicates statistical significance (*p* < 0.05).

**Figure 8 sensors-24-03331-f008:**
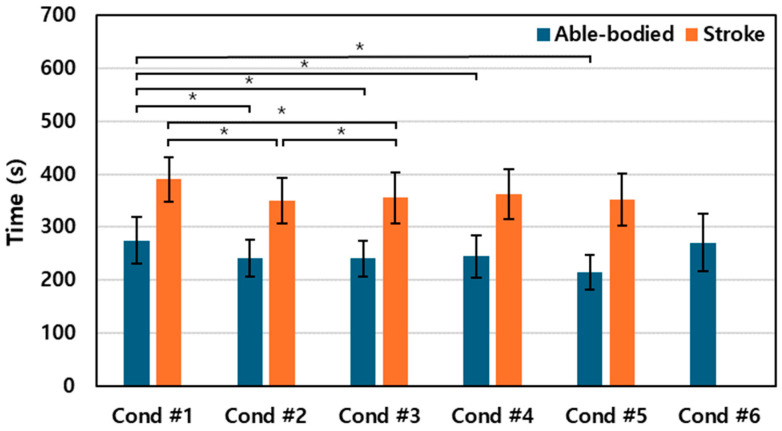
Movement time for able-bodied and stroke groups. The asterisk indicates statistical significance (*p* < 0.05).

**Figure 9 sensors-24-03331-f009:**
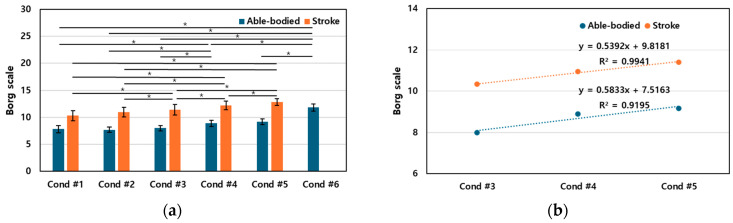
Borg scale for all participants. (**a**) The mean and standard deviation of the Borg scale. (**b**) The regression equation for the Borg scale from conditions #3 to #5 revealed that as the synchronization ratio of trunk movements increased, there was a proportional increase in the Borg scale. In the regression equation, Condition #3, Condition #4, and Condition #5 correspond to 1, 2, and 3, respectively. The asterisk indicates statistical significance (*p* < 0.05).

**Figure 10 sensors-24-03331-f010:**
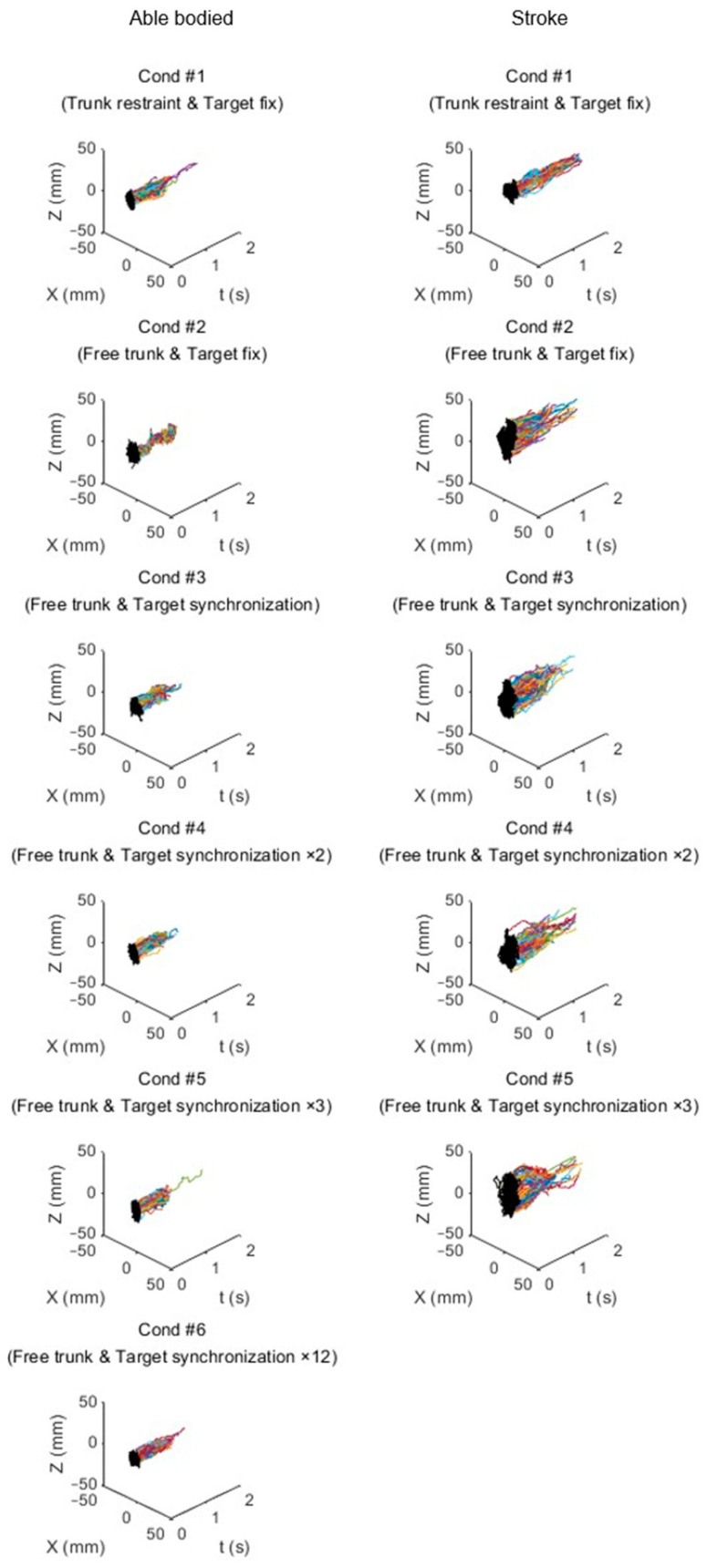
Trunk movement trajectory for an able-bodied participant (1st column) and a stroke participant (2nd column). The *Z*-axis represents the anterior–posterior direction, and the *X*-axis signifies the medial–lateral direction. Additionally, to capture the extent of trunk movement, the trajectories were projected onto the XZ plane and displayed together. The different colors in the figure represent the trajectories of arm movements based on the number of repetitions.

## Data Availability

The data presented in this study are available on request from the corresponding author.

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
