# Peer review of "Mitigating Trunk Compensatory Movements in Post-Stroke Survivors through Visual Feedback during Robotic-Assisted Arm Reaching Exercises"

_sensors, 2024, doi:10.3390/s24113331_

Round 1

Reviewer 1 Report

Comments and Suggestions for Authors

General: 

The topic of the current article is extremely useful as well the objective is relevant. 

Even so, I believe it has a few limitations, which I will enumerate below. 

.

This work can be accepted with minor revision.

1.    Abstract:

The abstract effectively communicates the objective of the study, and provides a succinct summary of the study design, methodology, and key findings. The findings regarding the effectiveness of visual feedback in reducing trunk compensatory movements are important for the field of rehabilitation and have practical implications for improving post-stroke rehabilitation strategies.

Implications: The abstract discusses the potential implications of the findings for robotic-assisted rehabilitation strategies, emphasizing the importance of customized visual feedback in correcting aberrant upper limb movements among stroke survivors.

However, the abstract could benefit from providing more context on the prevalence and impact of trunk compensatory movements in post-stroke rehabilitation. This would help readers understand the significance of the study and its relevance to current clinical practices.

The abstract mentions "customized visual feedback" without providing details on what this entails or how it was implemented in the study. Providing more specific information on the visual feedback modalities used in the study would enhance the clarity and transparency of the findings.

Overall, while the abstract effectively summarizes the main findings of the study, addressing the above critiques would improve the clarity, context, and relevance of the abstract for readers in the field of rehabilitation and stroke rehabilitation in particular.

2.    Keywords:

No specific comments

3.    Introduction

No specific comments .

4.Materials and Methods

It is unclear whether the study received approval from the ethics committee and what ethical procedures were implemented during data collection. Additionally, there is a lack of clarity regarding how the sample was stratified or the criteria for inclusion and exclusion. It is recommended that the authors incorporate this information into the article. Furthermore, it is proposed that a paragraph be added at the outset of the Materials and Methods section to outline the overall organization of the study.

Furthermore, while the authors stated that 18 stroke survivors participated in the study, they did not provide details on the representativeness of this sample or any potential limitations associated with the sample size.

The section provides a comprehensive overview of the NREH device, the methodology for detecting trunk motion, participant demographics, and the experimental setup. The description of the NREH device and its capabilities is clear and informative. Additionality, the integration of Microsoft’s Azure Kinect for trunk motion detection and the Unity framework for visual feedback is well-explained. The use of TCP/IP socket communication for data exchange is appropriate for real-time feedback.

The inclusion criteria for both able-bodied individuals and stroke survivors are clearly defined, ensuring relevance to the study objectives.

The description of the experimental setup, including participant positioning, target synchronization, and exercise categorization, is detailed and facilitates replication.

The study employs novel techniques for trunk motion detection and real-time visual feedback, contributing to the advancement of upper limb rehabilitation.

The ability to adjust exercise difficulty levels and provide targeted feedback highlights the versatility of the NREH device, catering to individual patient needs.

The section provides sufficient detail for replication and understanding, enhancing the study's credibility and potential for further research.

5. Results

The statistical analysis using SPSS and the Wilcoxon signed-rank test is appropriate for comparing trunk motion metrics across conditions. The presentation of results is clear, with figures and tables aiding comprehension.

Interpretation: The interpretation of trunk motion metrics, such as range of motion and position difference, provides valuable insights into the effectiveness of visual feedback in reducing compensatory movements.

The study evaluates multiple aspects of trunk motion, including range of motion, position difference, movement time, and perceived effort, providing a holistic understanding of the intervention's impact.

Figures and tables effectively summarize key findings, enhancing the readability and accessibility of the results section.

6. Discussion

The discussion of results in the context of stroke rehabilitation underscores the practical implications of the study findings for improving therapeutic outcomes.

7. Conclusions

The conclusion concisely summarizes the main findings of the study, highlighting the significant impact of visual feedback in reducing trunk compensation during robotic-assisted arm reaching exercises in stroke survivors. Identifies important areas for future research, such as exploring the influence of multimodal feedback and conducting longitudinal clinical trials to assess long-term effects.

It emphasizes the importance of developing evidence-based guidelines for integrating visual feedback into clinical practice, suggesting a commitment to advancing the field.

It underscores the need for further understanding of visual feedback in robotic rehabilitation to develop more targeted and effective strategies for stroke rehabilitation.

While the conclusion suggests the need for future research, it could benefit from more specific recommendations on the design and focus of such studies, as well as potential strategies for optimizing feedback modalities.

While the conclusion addresses the importance of developing evidence-based guidelines, it could further discuss the practical implications of the study's findings for clinicians and researchers in the field of stroke rehabilitation.

The conclusion mentions the importance of comparing feedback-based interventions to conventional approaches but does not provide clarity on how such comparisons could be conducted or what specific outcomes might be of interest.

Overall, the conclusion effectively summarizes the study's findings and provides valuable insights into future research directions. However, it could be strengthened by providing more specific recommendations and discussing the practical implications of the findings in more detail. Additionally, clarity on how comparative analyses could be conducted would enhance the conclusion's impact.

 8.    References

No specific comments

9.    Figures

No specific comments

Author Response

[Reviewer #1]

Comments and Suggestions for Authors

General: 

The topic of the current article is extremely useful as well the objective is relevant. 

Even so, I believe it has a few limitations, which I will enumerate below.  

This work can be accepted with minor revision.

  1. Abstract:

(Comment) The abstract effectively communicates the objective of the study, and provides a succinct summary of the study design, methodology, and key findings. The findings regarding the effectiveness of visual feedback in reducing trunk compensatory movements are important for the field of rehabilitation and have practical implications for improving post-stroke rehabilitation strategies.

Implications: The abstract discusses the potential implications of the findings for robotic-assisted rehabilitation strategies, emphasizing the importance of customized visual feedback in correcting aberrant upper limb movements among stroke survivors.

However, the abstract could benefit from providing more context on the prevalence and impact of trunk compensatory movements in post-stroke rehabilitation. This would help readers understand the significance of the study and its relevance to current clinical practices.

The abstract mentions "customized visual feedback" without providing details on what this entails or how it was implemented in the study. Providing more specific information on the visual feedback modalities used in the study would enhance the clarity and transparency of the findings.

Overall, while the abstract effectively summarizes the main findings of the study, addressing the above critiques would improve the clarity, context, and relevance of the abstract for readers in the field of rehabilitation and stroke rehabilitation in particular.

(Response) Thank you for your valuable comments and suggestions regarding our abstract. We have carefully revised the abstract to address your critiques and improve its overall quality.

  1. To provide more context on the prevalence and impact of trunk compensatory movements in post-stroke rehabilitation, we have added the following statement: "These aberrant movements are prevalent among stroke survivors and can hinder their progress in rehabilitation, making it crucial to address this issue.” This addition highlights the significance of our study and its relevance to current clinical practices in stroke rehabilitation.
  2. We have also clarified the specific visual feedback modalities used in the study by stating, " In the visual feedback modalities, the target position was synchronized with trunk movement at ratios where the target moved at the same speed, double, and triple the trunk's motion speed, providing real-time feedback to the participants.” This additional detail provides a clearer understanding of how the visual feedback was implemented and customized in the study.

We believe that these revisions have significantly enhanced the clarity, context, and relevance of our abstract, allowing readers to better grasp the importance of our study and its potential implications for improving post-stroke rehabilitation strategies.

  1. Keywords:

No specific comments

  1. Introduction

No specific comments.

4.Materials and Methods

(Comment) It is unclear whether the study received approval from the ethics committee and what ethical procedures were implemented during data collection.

(Response) Thank you for your comments regarding the clarity of ethical procedures in our manuscript. We value your attention to the ethical dimensions of our research.

In response to your feedback, we have revised the Materials and Methods section of our manuscript to clearly detail the ethical approvals and procedures associated with our study. Specifically, we have added explicit confirmation that the study was approved by the Institutional Review Board (IRB) of the National Rehabilitation Center, under the protocol number NRC-2022-07-057. Additionally, we have emphasized that all participants in the study provided written informed consent before participating. This information has been prominently placed at the beginning of the Participants subsection to ensure it is easily accessible and to highlight our commitment to ethical research practices.

We hope that these revisions address your concerns and provide the necessary transparency about how we conducted the research in an ethically rigorous manner. Thank you for helping us improve the integrity and quality of our manuscript.

(Comment) Additionally, there is a lack of clarity regarding how the sample was stratified or the criteria for inclusion and exclusion. It is recommended that the authors incorporate this information into the article.

(Response) Thank you for your constructive feedback regarding the clarity of the sample stratification and the inclusion and exclusion criteria in our study. We appreciate the opportunity to enhance the transparency and comprehensiveness of our manuscript.

In response to your comment, we have revised the Materials and Methods section to clearly delineate how the sample was stratified and to specify the criteria for participant inclusion and exclusion. These changes aim to provide readers with a thorough understanding of our study's methodology and the demographic and clinical profile of our participants.

  1. Stratification and Sampling Method: The study participants were categorized into two groups: able-bodied individuals and stroke survivors with impaired reaching abilities. The sample was stratified based on the presence of neurological conditions affecting upper limb function. Participants were recruited through convenience sampling at the National Rehabilitation Center, and the sample size was determined based on precedents set by similar studies in the field, ensuring statistical power and relevance.
  2. Inclusion and Exclusion Criteria: For stroke survivors, inclusion criteria included being aged 19 years or older, diagnosed with unilateral stroke resulting in hemiplegia, at least six months post-stroke, having a Medical Research Council (MRC) shoulder score of 2 or higher, a Modified Ashworth Scale (MAS) score of less than 3, a Line Bisection Test (LBT) result of less than 12.5 mm, and a Korean Mini-Mental Status Examination (K-MMSE) score of over 18. Exclusion criteria were visual field defects or reduced visual acuity, other neurological or orthopedic problems in the upper limbs, current medication for pain, or severe contracture in the elbow joint.
  3. Demographic and Clinical Characteristics: We have also included detailed demographic and clinical characteristics of the participants, such as the average duration since stroke onset, functional assessment scores (Fugl-Meyer Assessment, Box and Block Test, Action Research Arm Test, and Modified Barthel Index), which provide insight into the participants' rehabilitation potential.

               We hope these revisions address your concerns effectively and clarify the methodological rigor and ethical considerations of our study. We are grateful for your guidance in improving the quality and integrity of our work.

(Comment) Furthermore, it is proposed that a paragraph be added at the outset of the Materials and Methods section to outline the overall organization of the study. 

(Response) Thank you for your suggestion to enhance the clarity of our manuscript by providing an outline at the beginning of the Materials and Methods section. In response to your comment, we have added a new paragraph at the start of Section 2. This paragraph now outlines the overall organization of the study, detailing each key component and its role within the research framework. This addition will help readers navigate through our methodology more effectively and understand the systematic approach we employed in conducting our study. We believe this enhancement improves the structural flow and comprehensibility of our manuscript.

(Comment) Furthermore, while the authors stated that 18 stroke survivors participated in the study, they did not provide details on the representativeness of this sample or any potential limitations associated with the sample size.

(Response) Thank you for your insightful comments regarding the sample size and the representativeness of the stroke survivor group in our study. We appreciate the opportunity to enhance the clarity and depth of our manuscript with this vital information.

In response to your feedback, we have revised the Participants subsection to provide a more comprehensive analysis of the representativeness of our stroke survivor sample and the potential limitations related to the sample size. Specifically, we have acknowledged that the sample size of 18 stroke survivors, while consistent with similar studies in the field, such as those conducted by Sivan et al. (2014) and Masiero et al. (2007), may still limit the generalizability of our findings to the broader stroke population. This potential limitation is crucial for interpreting the applicability of our results.

Additionally, we have addressed the use of convenience sampling for participant recruitment at the National Rehabilitation Center, noting that this method could introduce selection bias, which might affect the diversity and representativeness of the sample. We have discussed these aspects further in the revised Discussion section to ensure that readers are aware of these critical methodological considerations and their implications for the study’s findings.

These revisions aim to provide readers with a transparent overview of how the sample was chosen and its potential impact on the study's conclusions, thereby enhancing the manuscript's scientific rigor and credibility. We hope these updates satisfactorily address your concerns and improve the overall quality of our work.

  1. Results

(Comment) The statistical analysis using SPSS and the Wilcoxon signed-rank test is appropriate for comparing trunk motion metrics across conditions. The presentation of results is clear, with figures and tables aiding comprehension.

Interpretation: The interpretation of trunk motion metrics, such as range of motion and position difference, provides valuable insights into the effectiveness of visual feedback in reducing compensatory movements.

The study evaluates multiple aspects of trunk motion, including range of motion, position difference, movement time, and perceived effort, providing a holistic understanding of the intervention's impact.

Figures and tables effectively summarize key findings, enhancing the readability and accessibility of the results section.

(Response) Thank you for your positive feedback.

  1. Discussion

(Comment) The discussion of results in the context of stroke rehabilitation underscores the practical implications of the study findings for improving therapeutic outcomes.

(Response) Thank you for your positive feedback.

  1. Conclusions

The conclusion concisely summarizes the main findings of the study, highlighting the significant impact of visual feedback in reducing trunk compensation during robotic-assisted arm reaching exercises in stroke survivors. Identifies important areas for future research, such as exploring the influence of multimodal feedback and conducting longitudinal clinical trials to assess long-term effects.

It emphasizes the importance of developing evidence-based guidelines for integrating visual feedback into clinical practice, suggesting a commitment to advancing the field.

It underscores the need for further understanding of visual feedback in robotic rehabilitation to develop more targeted and effective strategies for stroke rehabilitation.

(Comment) While the conclusion suggests the need for future research, it could benefit from more specific recommendations on the design and focus of such studies, as well as potential strategies for optimizing feedback modalities.

(Response) Thank you for your comment on the study's conclusion. We agree that discussing the practical implications for clinicians and researchers is crucial. In response, we have emphasized that future research should focus on optimizing feedback modalities and evaluating their long-term efficacy. This includes conducting randomized controlled trials to compare visual feedback interventions with conventional therapy methods. These studies will help develop evidence-based guidelines that consider practical aspects such as session frequency, duration, and customization to patient needs, ultimately aiding in the integration of visual feedback strategies into clinical practice for enhancing stroke rehabilitation outcomes.

  1. References

No specific comments

  1. Figures

No specific comments

Reviewer 2 Report

Comments and Suggestions for Authors

The manuscript investigated the effect of visual feedback on reducing trunk compensatory during robotic-assisted arm-reaching exercises. A comparison was made between healthy subjects and stroke patients in the Trunk Range of Motion, the Trunk Position Difference, the Movement Time, the Borg Scale, and the Trajectory when the participants performed a reaching task under 6 conditions in three directions (medial, contralateral, and ipsilateral). Results showed applying target movements ranging from 1 to 2 times the trunk movement is deemed reasonable for stroke participants.

There are a few comments that need to be addressed in the revision:

Line 194~ Line 204: The order of target directions was introduced. Whether participants perform sequentially from Condition#1 to Condition#6? Whether sequential effects affect the results of the experiment?

Line 206~ Line 232: Whether it more appropriate to put them in “2. Materials and Methods” as “Data Analysis and Statistical Analysis” instead of in "3. Results"?

Line 209~ Line 213: Whether it more appropriate to put the information about the subjects taking part in the experiment in “2.3. Participants”?

Line 378~ Line 387: Whether it more appropriate to put the position data filtering process in “2. Materials and Methods”?

Equation (2), (3), (4), and (5) have two P_target, if they should be distinguished?

Figures (6), (7), and (8) don't have any significance markers.

The discussion section needs to further explain the reasons for the findings, for example, based on sensory-motor mechanisms or error-based learning theories, etc.

The present study compared the effects of different incremental ratios of visual feedback on trunk compensation. However, whether the changes in the evaluation metrics were due to transient effects of the feedback modality or actual motor skills learning needs to be further explored. Future experiments should be better designed to evaluate stroke patients’ functional improvements and the retention and generalization ability of different visual feedback training.

Author Response

Reviewer #2

Comments and Suggestions for Authors

The manuscript investigated the effect of visual feedback on reducing trunk compensatory during robotic-assisted arm-reaching exercises. A comparison was made between healthy subjects and stroke patients in the Trunk Range of Motion, the Trunk Position Difference, the Movement Time, the Borg Scale, and the Trajectory when the participants performed a reaching task under 6 conditions in three directions (medial, contralateral, and ipsilateral). Results showed applying target movements ranging from 1 to 2 times the trunk movement is deemed reasonable for stroke participants.

There are a few comments that need to be addressed in the revision:

(Comment) Line 194~ Line 204: The order of target directions was introduced. Whether participants perform sequentially from Condition#1 to Condition#6? Whether sequential effects affect the results of the experiment?

(Response) Thank you for your insightful comments regarding the potential influence of sequential effects due to the order of target directions in our experimental setup. We acknowledge the importance of addressing these concerns to ensure the validity of our findings.

In response to your observations, we have incorporated specific measures into our methodology section to explicitly detail the strategies we employed to mitigate such effects:

  1. Adequate Rest Periods: We provided participants with sufficient rest periods between conditions to ensure they had ample time to recover and minimize any carry-over effects from one condition to another. These rest periods allowed participants to maintain their focus and performance throughout the experiment, thus reducing the impact of fatigue or habituation.
  2. Progressive Task Difficulty: The tasks were presented in an order of increasing difficulty, starting with the least challenging task (Condition #1) and progressing to more demanding tasks (up to Condition #6). This approach was chosen to maintain participants' motivation and engagement throughout the experiment. By starting with the easiest task, participants could familiarize themselves with the experimental setup and gradually adapt to the increasing complexity of the tasks, minimizing the potential for frustration or discouragement.

These measures were implemented to ensure that any potential learning or fatigue effects did not unduly influence the results. By structuring the conditions to gradually increase in difficulty and interspersing them with adequate rest periods, we aimed to counteract potential biases that might arise from participants becoming either increasingly proficient due to practice or fatigued over time. The manuscript has been updated to include a detailed explanation of these methodological considerations to provide clarity and enhance understanding of the study's design and the validity of its conclusions.

(Comment) Line 206~ Line 232: Whether it more appropriate to put them in “2. Materials and Methods” as “Data Analysis and Statistical Analysis” instead of in "3. Results"?

(Response) We appreciate the reviewer’s suggestion and agree that the placement of the data analysis and statistical analysis sections would be more appropriately situated within "2. Materials and Methods." Accordingly, these sections have been moved to subsection 2.5, titled “Data Analysis and Statistical Analysis.” This change enhances the logical flow of the manuscript by aligning the methods employed directly with the description of materials and procedures.

(Comment) Line 209~ Line 213: Whether it more appropriate to put the information about the subjects taking part in the experiment in “2.3. Participants”?

(Response) Thank you for your constructive comment. We have relocated the information regarding the subjects involved in the experiment to subsection "2.4. Participants." Additionally, the section previously labeled "2.4. Experimental Setup" has been renumbered to "2.3. Experimental Setup" to maintain the coherence and sequential order of the manuscript sections. This adjustment ensures that the description of the participants directly precedes the experimental setup, reflecting their chronological order in the study methodology.

(Comment) Line 378~ Line 387: Whether it more appropriate to put the position data filtering process in “2. Materials and Methods”?

(Response) Thank you for your suggestion. Upon review, we concur that the description of the position data filtering process is more aptly classified under the "Materials and Methods" section. It has now been moved to subsection "2.2. Detecting Trunk Motion via RGB-D Camera." This reorganization ensures that the methodology is comprehensively detailed in the appropriate section, thereby improving the clarity and logical flow of the paper.

(Comment) Equation (2), (3), (4), and (5) have two P_target, if they should be distinguished?

(Response) Thank you for highlighting the use of P_target in equations (2), (3), (4), and (5). To clarify, the P_target on the right-hand side of each equation represents the initial position, and the P_target on the left is the updated position after adjustments. This notation shows the dynamic updating of the target based on changes in trunk position, multiplied by different speed factors. We have updated the manuscript to better explain this calculation process and ensure clear understanding. We now specify that: "In Equations (2), (3), (4), and (5), P_target on the left-hand side represents the updated target position, while the P_target on the right-hand side refers to its prior value. The target position is updated by adding the change in trunk position (ΔP_Trunk), scaled by the respective speed factor (1, 2, 3, or 12), to the current target position. This calculation reflects the dynamic adaptation of target placement in response to the participant's trunk movements, simulating real-world adjustments and enhancing task complexity."

(Comment) Figures (6), (7), and (8) don't have any significance markers.

(Response) Thank you for pointing out the absence of significance markers in Figures 6, 7, and 8. We have updated these figures to include significance markers to better illustrate the statistical differences observed in our study. This should enhance the clarity and interpretability of the results presented.

(Comment) The discussion section needs to further explain the reasons for the findings, for example, based on sensory-motor mechanisms or error-based learning theories, etc.

(Response) Thank you for your comment highlighting the need for a more in-depth explanation of the mechanisms behind our findings. We have updated the discussion section to better reflect how sensory-motor learning and error-based learning theories support the observed effects of visual feedback in our study. By incorporating real-time visual feedback, the system enables stroke survivors to immediately recognize and correct deviations in trunk movements. This approach aligns with error-based learning, where discrepancies between intended and actual movements are corrected, thus enhancing motor learning and minimizing compensatory behaviors. Additionally, we discussed how this method improves sensory feedback, which is crucial for reinforcing correct motor patterns and strengthening the trunk, further contributing to the efficacy of rehabilitation exercises.

(Comment) The present study compared the effects of different incremental ratios of visual feedback on trunk compensation. However, whether the changes in the evaluation metrics were due to transient effects of the feedback modality or actual motor skills learning needs to be further explored. Future experiments should be better designed to evaluate stroke patients’ functional improvements and the retention and generalization ability of different visual feedback training.

(Response) Thank you for your insightful comment. We acknowledge the need to determine whether the observed improvements are due to temporary effects of visual feedback or actual learning of motor skills. In response, we have refined our future research plans outlined in the manuscript. We will design experiments to more effectively evaluate not only the immediate functional improvements in stroke patients but also the long-term retention and generalization of skills acquired through different visual feedback training regimes. This will involve structured follow-up assessments to monitor the sustainability of the intervention effects, thereby ensuring that our findings contribute meaningfully to the field of stroke rehabilitation.